# The First Report on the Application of ISSR Markers in Genetic Variance Detection among Butterfly Pea (*Clitoria ternatea* L.) Accession in North Maluku Province, Indonesia

Nurhasanah [1,2,*], Reginawanti Hindersah [3], Tarkus Suganda [4], Vergel Concibido [5], Sundari [2,*] and Agung Karuniawan [6,*]

1 Doctoral Program of Agricultural Science, Faculty of Agriculture, Universitas Padjadjaran, Sumedang 45363, Indonesia
2 Department of Biology Education, Faculty of Teacher Training and Education, Universitas Khairun, Ternate 97728, Indonesia
3 Department of Soil, Faculty of Agriculture, Universitas Padjadjaran, Sumedang 45363, Indonesia; reginawanti@unpad.ac.id
4 Department of Plant Pest and Disease, Faculty of Agriculture, Universitas Padjadjaran, Sumedang 45363, Indonesia; tarkus.suganda@unpad.ac.id
5 Sensient Technologies Corporation, 777 E Wisconsin Ave, Milwaukee, WI 53202, USA; vergel.concibido@sensient.com
6 Department of Agronomy, Faculty of Agriculture, Universitas Padjadjaran, Sumedang 45363, Indonesia
* Correspondence: nurhasanah19003@mail.unpad.ac.id (N.); sundari@unkhair.ac.id (S.); agung.karuniawan@unpad.ac.id (A.K.); Tel.: +62-82190515858 (S.); +62-81214694288 (A.K.)

**Abstract:** Butterfly pea (*Clitoria ternatea* L.) is a leguminous plant with several potential health benefits. The scientific name is derived from its origin on Ternate Island, North Maluku. Therefore, this study aimed to analyze the genetic variability in butterfly pea using Intergenic Simple Sequence Repeat (ISSR) markers in North Maluku. Field surveys, collection trips, and habitat studies of butterfly pea plants were conducted on Ternate, Tidore, Halmahera, and Morotai Islands. Genetic diversity was analyzed based on molecular data from the ISSR method. The molecular analysis results obtained using PCR-ISSR on 18 accessions showed a low degree of similarity. Among these, 15 accessions from Ternate, Tidore, Morotai, and Halmahera were in group A, while 3 from Ternate, Tidore, and Halmahera were in group B. All accessions exhibited a genetic similarity level of 0.709, indicating significant diversity. The arrangement among accessions on the dendrogram was similar to the phylogenetic tree, showing separation and spread at 0.608–0.924 based on the Jaccard coefficient. The results suggested that *C. ternatea* probably originated from Ternate, and subsequently spread to Tidore, Halmahera, and Morotai due to its use as a herbal medicine and ornamental plant. This information could be used as the basis for butterfly pea conservation and cultivation activities in Indonesia, specifically in Ternate Island, North Maluku.

**Keywords:** Clitoria; Ternate; butterfly pea; genetic variance; ISSR

## 1. Introduction

Butterfly pea (*Clitoria ternatea* L.) is a leguminous plant from the Fabaceae family which is native to Asia and popular for its bright blue flowers. The plant is rich in antioxidants and often used in making herbal tea as well as a natural dye. The global geographical distribution encompasses the Indian Ocean, including Maluku. However, data on plant diversity in this region have not been documented by the *Center for Agriculture and Biosciences International* (CABI) [1]. North Maluku is one of the islands in Eastern Indonesia, *constituting part* of the Maluku Archipelago [1]. The species marker "ternatea" is presumably derived from the name of Ternate Island, as indicated by previous studies [2–5]. Additional data on genetic diversity can be interesting because they can potentially provide information

on the genetic value of the various accessions used in breeding and crop improvement. They can also provide a glimpse of the domestication of the species. More importantly, they can provide the basis for future genomic endeavors like the creation of a reference genome map that will facilitate marker development at the very least and, ultimately, gene editing [6]. Information related to the genetic variability in butterfly pea, specifically in Maluku, remains limited. In a previous study, flower morphology characters and yield components were used to determine the genetic diversity of 10 accessions across specimens obtained from Java and Sumatra in a uniform environment [7]. A description of the morphological variations was also carried out on 26 accessions from Bali, with the results showing diversity in petal color, crown, and stamen structure [8]. Morphological variability can be assessed through a molecular analysis method using markers. The RAPD molecular marker was used to determine the genetic diversity and relatedness of four *Clitoria* genotypes in India, resulting in four distinct groups [9]. This marker has also been employed to assess the genetic variability in butterfly pea populations [10]. However, genetic diversity data for North Maluku are currently lacking. According to previous studies, genetic variability plays a crucial role in plant breeding and use [11]. The available data on the diversity of Indonesian butterfly pea could enhance the existing genetic materials, specifically for selecting the desired parents. Developing germplasm from local accessions in North Maluku holds significant potential for this purpose. This study was conducted to determine genetic diversity using molecular characteristics through Intergenic Simple Sequence Repeat (ISSR) markers across four islands. The use of molecular markers has become a standard method to study variability among closely related taxa [12]. Genetic markers, such as isozymes [13] and polymerase chain reaction (PCR)-based methods, are more reliable for the identification of genetic diversity compared to morphological markers, although each technique has advantages and limitations. ISSR is a PCR-based method, which involves the amplification of DNA segments present at an amplifiable distance between two identical microsatellite repeat regions oriented in opposite directions. The technique uses microsatellites, usually 16–25 bp long, as primers in a single-primer PCR reaction targeting multiple genomic loci to amplify mainly ISSRs of different sizes. The ISSR method is similar to RAPD since both require no previous knowledge of the genome, cloning, or specific primer design, but it has higher reproducibility due to the use of higher annealing temperatures. Additionally, the cost of ISSR analyses is lower compared to AFLP. This has led to the broad use of ISSR across various studies on genetic diversity, phylogenetics, genetic mapping, and evolutionary biology in a wide range of plant species [14]. This study aimed to evaluate the genetic variability in and relatedness of 18 accessions in butterfly pea from North Maluku province using 10 ISSR primers. At this time, there is no DNA sequence database from the butterfly pea plant from Ternate Island and the North Maluku Islands at GeneBank (NCBI). It is hoped that this research information will provide initial data on which accessions need to be sequenced as representative accessions.

## 2. Materials and Methods

### 2.1. Study Area and Sample Collection

The plant sampling areas are shown in Figure 1, and data were collected across 4 islands, namely Ternate, Tidore, Halmahera, and Morotai, from February 2020 to September 2021.

Plant sampling was carried out randomly using roaming techniques and assisted by local people familiar with butterfly pea. The coordinates and altitude of the location, as well as environmental habitat characteristics, were also measured (Table 1).

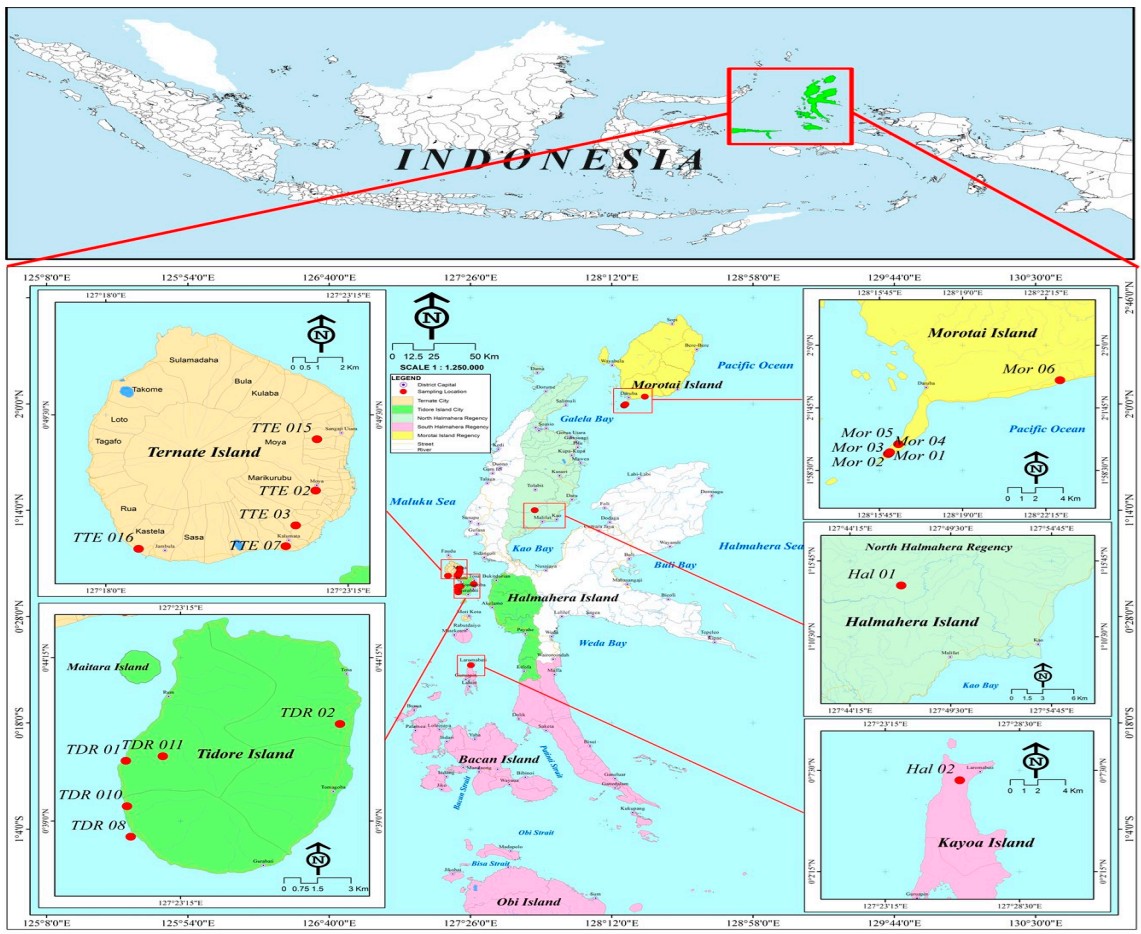

**Figure 1.** The study area of butterfly pea in North Maluku.

**Table 1.** Geographical coordinates of 18 accessions of butterfly pea.

| Districts | Sum of Accessions | Accession Codes | Longitude E | Latitude N |
|---|---|---|---|---|
| North Halmahera | 1 | Hal 01 | 127°46′55.513″ | 1°14′03.224″ |
| South Halmahera | 1 | Hal 01 | 127°26′10.100″ | 0°06′59.400″ |
| South Morotai | 6 | Mor 01 | 128°16′05.502″ | 1°59′21.498″ |
| | | Mor 02 | 128°16′08.003″ | 1°59′24.420″ |
| | | Mor 03 | 128°16′07.776″ | 1°59′24.156″ |
| | | Mor 04 | 128°16′09.174″ | 1°59′26.736″ |
| | | Mor 05 | 128°16′29.682″ | 1°59′52.223″ |
| | | Mor 06 | 128°22′48.263″ | 2°3′11.193″ |
| North Tidore | 5 | TDR 01 | 127°21′55.741″ | 0°40′56.199″ |
| East Tidore | | TDR 02 | 127°27′06.416″ | 0°42′07.130″ |
| South Tidore | | TDR 08 | 127°22′2.755″ | 0°38′29.887″ |
| | | TDR 010 | 127°21′57.475″ | 0°39′28.701″ |
| North Tidore | | TDR 011 | 127°22′49.284″ | 0°41′0,5.041″ |
| Central Ternate | 5 | TTE 02 | 127°22′32.977″ | 0°47′19.263″ |
| South Ternate | | TTE 03 | 127°22′07.081″ | 0°46′18.963″ |
| | | TTE 07 | 127°21′54.190″ | 0°45′43.103″ |
| North Ternate | | TTE 015 | 127°22′34.723″ | 0°48′47.968″ |
| Ternate Island | | TTE 016 | 127°18′42.710″ | 0°45′38.461″ |

Note: Hal = Halmahera Island; Mor = Morotai Island, TDR = Tidore Island, and TTE = Ternate Island.

## 2.2. Plant Material

### 2.2.1. Sampling of Plant Material

All accessions were selected based on the characterization of plant parts as outlined in flora From Panama [15]. These accessions served as representative samples for estimating the existing genetic diversity (Figure 2).

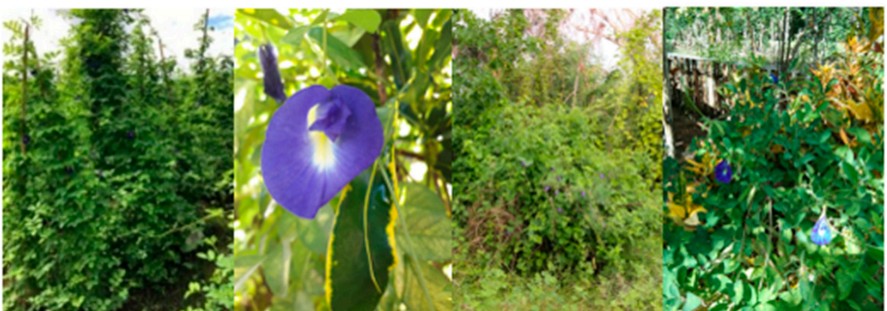

**Figure 2.** Populations of butterfly pea found on the Ternate, Halmahera, Morotai, and Tidore Islands in the North Maluku Archipelago.

### 2.2.2. Total DNA Isolation

DNA isolation and ISSR analysis were carried out using fresh leaf samples of 18 accessions collected across four islands in North Maluku. The extraction was performed using the Wizard® Genomic DNA Purification Kit (Promega, Madison, WI, USA), while isolation results were tested qualitatively through electrophoresis with 1% agarose gel.

### 2.2.3. PCR Amplification

Molecular diversity determination was conducted using ISSR molecular markers, and the primers consisted of 10 sets, in line with a previous study [13]. Primers included (AG)8G, (CA)6GG, 25CC, (AC)8C, (AG)8C, (CT)8RC, (GAA)6, UBC 887, UBC 889, and GT 8T. As described in Table 2, the amplification reaction was performed using a PCR machine, with each 10 μL reaction containing 5 μL of PCR Master MyTaq HS Red Mix, 1 μL of template DNA sample (100 ng/μL), 2 μL of water, and 1 μL of primer (10 pmol). The process was carried out with the following temperature settings: predenaturation at 94 °C for 4 min, 36 cycles consisting of denaturation at 94 °C for 2 min, annealing for 1 min, and extension at 72 °C for 2 min, while the postelongation occurred at 72 °C for 10 min. Furthermore, DNA bands from the PCR results were electrophoresed using 2% agarose gel and visualized under a UV transilluminator.

**Table 2.** ISSR primer list and condition of PCR, polymorphic and monomorphic bands, and percentage of polymorphic analysis of 18 butterfly pea accessions.

| No | Primers | Sequence (5′-3′) | Tm (°C) | TB | PB | MB | P | %M | PIC |
|----|---------|------------------|---------|----|----|----|----|----|-----|
| 1 | (AG)8 | AGAGAGAGAGAGAGAGG | 54.5 | 8 | 7 | 1 | 88 | 12 | 0.35 |
| 2 | (CA)6GG | CACACACACACAGG | 42.0 | 6 | 5 | 1 | 83 | 17 | 0.40 |
| 3 | 25CC | AGGGCTGGAGGAGGGC | 50.0 | 6 | 6 | 0 | 100 | 0 | 0.33 |
| 4 | (AC)8C | ACACACACACACACACC | 52.0 | 6 | 6 | 0 | 100 | 0 | 0.21 |
| 5 | (AG)8C | AGAGAGAGAGAGAGAGC | 52.0 | 6 | 3 | 3 | 50 | 50 | 0.54 |
| 6 | (CT)8RC | CTCTCTCTCTCTCTCTRC | 54.0 | 7 | 7 | 0 | 100 | 0 | 0.34 |
| 7 | (GAA)6 | GAAGAAGAAGAAGAAGAA | 50.0 | 6 | 6 | 0 | 100 | 0 | 0.33 |
| 8 | UBC 887 | AGTACGAGTTCTCTCTCTCTC | 55.0 | 7 | 7 | 0 | 100 | 0 | 0.31 |
| 9 | UBC 889 | ACTCGTAGTACACACACACAC | 60.0 | 7 | 6 | 1 | 86 | 14 | 0.35 |
| 10 | (GT) 8T | GTGTGTGTGTGTGTGTT | 54 | 6 | 6 | 0 | 100 | 0 | 0.23 |
| | | Total | | 65 | 59 | 6 | 91 | 9 | 0.34 |

Description: TB = total bands; PB = polymorphic band; MB = monomorphic band; %P = polymorphic percentage; %M = monomorphic percentage; PIC = polymorphic information content.

### 2.2.4. ISSR-PCR Analysis

The PCR-ISSR results were analyzed descriptively by scoring the presence or absence of bands. Scores 1 and 0 indicated the presence and absence of band formation, respectively. Only strong and reproducible ISSR bands were scored, and the different patterns observed were recorded as discrete variables, using 1 and 0 to indicate the presence or absence of a unique pattern. Furthermore, relationships among individuals were determined using the distance matrix method. Nei and Li's Dice similarity coefficients were calculated for all pair-wise comparisons between individual samples to provide a distance matrix. A dendrogram was constructed from this matrix through a hierarchical cluster analysis based on the Unweighted Pair Group Method Algorithm (UPGMA) [16]. Binary data processing was then analyzed using the Ntsys program version 2.11x.

## 3. Results

*Molecular Analysis of Butterfly Pea Genetic Diversity Using ISSR Markers*

A total of 18 accessions from butterfly pea were analyzed molecularly using ISSR markers to determine the genetic diversity. The accessions used were 5, 5, 6, and 2 from Ternate, Tidore, Morotai, and Halmahera, respectively. To analyze the genetic diversity among the accessions, 10 ISSR primers were used. The total bands produced were 65, consisting of 59 polymorphic and 6 monomorphic, with an average polymorphic percentage of 91%. The largest polymorphic information content (PIC) of 0.54 was identified in the primer (AG)8C. Furthermore, primary base pair sizing detected alleles ranging from 100 to 1500 bp, and the total bands produced by each of the 10 ISSR primers were between 6 and 8 (Table 2).

ISSR markers were used to determine the diversity of accessions, and the six primers employed were 25CC, (AC)8C (AG)8C, (CT)8RC, (GAA)6, UBC 887, and GT 8T at 100% polymorphic value. Primer (AG)8G produced the highest number of band sizes, namely eight, while (AG)8C and (AC)8C yielded the largest PIC value and polymorphic percentage (Figure 3).

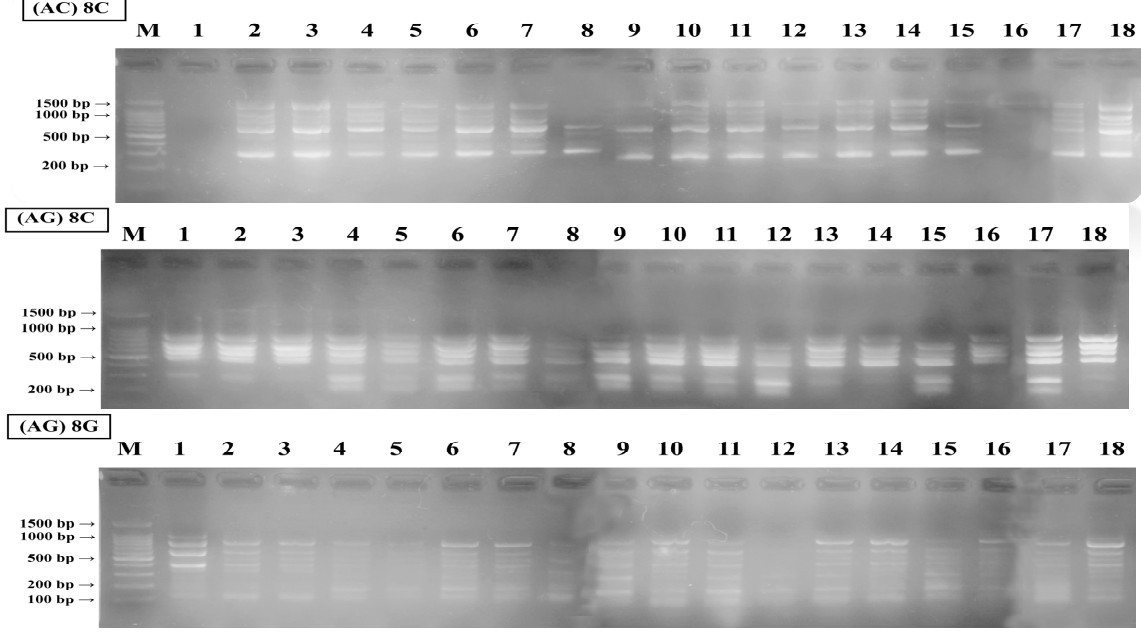

**Figure 3.** ISSR profile of selected primers. 1 = HAL 01; 2 = MOR 06; 3 = HAL 02; 4 = TDR 011; 5 = TDR 010; 6 = TDR 08; 7 = TDR 01; 8 = TDR 02; 9 = MOR 02; 10 = MOR 03; 11 = MOR 04; 12 = MOR 01; 13 = MOR 05; 14 = TTE 07; 15 = TTE 016; 16 = TTE 02; 17 = TTE 015; 18 = TTE 03; M = ISSR markers. HAL: Halmahera; MOR: Morotai; TDR: Tidore; and TTE: Ternate accession.

The cluster analysis of 18 butterfly pea accessions formed groups A and B; then, A was divided into subgroups A1 and A2, where A1 consisted of 3, 4, and 1 accession from Ternate, Morotai, and Tidore. Meanwhile, A2 had one, two, three, and one accession from Ternate, Morotai, Tidore, and Halmahera (HAL 02), respectively. Cluster B consisted of three accessions from Ternate, Tidore, and Halmahera (HAL 01) (Figure 4).

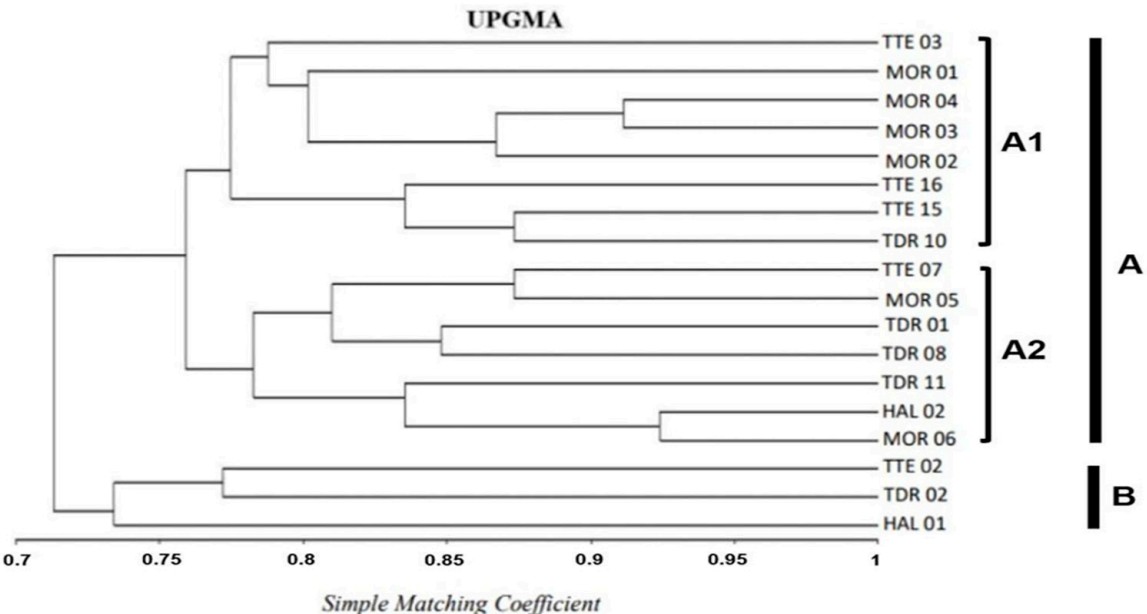

**Figure 4.** Grouping dendrogram of 18 butterfly pea accessions from North Maluku by ISSR PCR using the UPGMA method.

The Jaccard coefficient was used to determine the similarity of characteristics among the 18 accessions from grouping analysis based on the UPGMA method. The coefficient values observed ranged from 0.608 to 0.924. The highest percentage of genetic similarity was found between HAL 02 and MOR 06 at 92%, while MOR 01 and HAL 01 had the lowest at 60% (Table 3).

**Table 3.** Genetic similarity matrix using the Jaccard coefficient for butterfly pea accession.

| | HAL 01 | MOR 06 | HAL 02 | TDR 11 | TDR 10 | TDR 08 | TDR 01 | TDR 02 | MOR 02 | MOR 03 | MOR 04 | MOR 01 | MOR 05 | TTE 07 | TTE 016 | TTE 02 | TTE 015 | TTE 03 |
|---|---|---|---|---|---|---|---|---|---|---|---|---|---|---|---|---|---|---|
| HAL 01 | 1 | | | | | | | | | | | | | | | | | |
| MOR 06 | 0.823 | 1 | | | | | | | | | | | | | | | | |
| HAL 02 | 0.797 | 0.924 | 1 | | | | | | | | | | | | | | | |
| TDR 11 | 0.696 | 0.848 | 0.823 | 1 | | | | | | | | | | | | | | |
| TDR 10 | 0.709 | 0.785 | 0.759 | 0.861 | 1 | | | | | | | | | | | | | |
| TDR 08 | 0.671 | 0.747 | 0.722 | 0.722 | 0.861 | 1 | | | | | | | | | | | | |
| TDR 01 | 0.722 | 0.797 | 0.797 | 0.797 | 0.759 | 0.848 | 1 | | | | | | | | | | | |
| TDR 02 | 0.734 | 0.759 | 0.709 | 0.785 | 0.722 | 0.658 | 0.759 | 1 | | | | | | | | | | |
| MOR 02 | 0.684 | 0.759 | 0.734 | 0.835 | 0.797 | 0.684 | 0.785 | 0.823 | 1 | | | | | | | | | |
| MOR 03 | 0.684 | 0.785 | 0.81 | 0.81 | 0.797 | 0.759 | 0.81 | 0.747 | 0.848 | 1 | | | | | | | | |
| MOR 04 | 0.62 | 0.722 | 0.722 | 0.772 | 0.785 | 0.747 | 0.772 | 0.759 | 0.886 | 0.911 | 1 | | | | | | | |
| MOR 01 | 0.608 | 0.633 | 0.658 | 0.734 | 0.747 | 0.684 | 0.684 | 0.772 | 0.772 | 0.797 | 0.835 | 1 | | | | | | |
| MOR 05 | 0.696 | 0.772 | 0.772 | 0.772 | 0.759 | 0.772 | 0.823 | 0.709 | 0.810 | 0.861 | 0.848 | 0.759 | 1 | | | | | |
| TTE 07 | 0.722 | 0.823 | 0.823 | 0.797 | 0.785 | 0.823 | 0.823 | 0.684 | 0.734 | 0.835 | 0.772 | 0.658 | 0.873 | 1 | | | | |
| TTE 016 | 0.709 | 0.759 | 0.734 | 0.785 | 0.823 | 0.785 | 0.759 | 0.747 | 0.772 | 0.797 | 0.759 | 0.772 | 0.734 | 0.785 | 1 | | | |
| TTE 02 | 0.734 | 0.734 | 0.658 | 0.734 | 0.722 | 0.658 | 0.684 | 0.772 | 0.772 | 0.671 | 0.709 | 0.671 | 0.709 | 0.633 | 0.772 | 1 | | |
| TTE 015 | 0.658 | 0.709 | 0.684 | 0.785 | 0.873 | 0.785 | 0.709 | 0.696 | 0.797 | 0.747 | 0.785 | 0.797 | 0.759 | 0.734 | 0.848 | 0.722 | 1 | |
| TTE 03 | 0.684 | 0.759 | 0.759 | 0.785 | 0.747 | 0.684 | 0.684 | 0.696 | 0.797 | 0.797 | 0.81 | 0.747 | 0.81 | 0.785 | 0.722 | 0.696 | 0.797 | 1 |

## 4. Discussion

Based on the results, ISSR markers were successfully used to assess genetic variation in several *C. ternatea* accessions from Ternate, Tidore, Halmahera, and Morotai. The clustering pattern produced by the ISSR marker system did not show significant variation. In both grouping patterns, A and B, each consisted of accessions originating from various islands. Group A contained subgroups A1 and A2 originating from Ternate, Morotai, and Tidore, while B comprised accessions from Ternate, Tidore, and Halmahera. These groupings showed distinct genetic relationships influenced by geographic distribution patterns. All accessions from Ternate were distributed across the groups, confirming their highest genetic variation and adaptability to geographical conditions. The similarity and diversity values were 9% (%M) and 91% (%P), respectively, while the information polymorphic content was 34% (%PIC). This value showcases the advantage of ISSR markers in detecting genetic variability.

The wide distribution of *C. ternatea* accessions across subgroups on the dendrogram was attributed to the movement of planting material due to its popularity as herbal medicine. It could also be due to the aesthetic value, which attracts home gardeners, resulting in genetic lineage similarity [13]. However, the accessions from the islands were often grouped together. This grouping was justified by their geographical proximity and validated based on the analysis of ISSR markers. This study also indicated higher genetic similarities among the eighteen accessions. The cluster analysis suggested that the separation of accessions was more influenced by their geographical location than the molecular character of ISSR markers. When assessing genetic diversity at the genotype level, ISSR proved more informative than RAPD, as observed in this study and previous reports [17]. ISSR markers showed moderate genetic similarity among accessions within the population, ranging from 60 to 92%. This differed from a previous study on the Indian butterfly pea population, which yielded a high genetic similarity. However, samples collected from various areas in North Maluku Islands showed very high genetic similarity, supporting the results of previous studies [18]. The narrow genetic base may be due to the self-pollinating behavior of the species, which causes restrictions on gene flow [10]. Indigenous species generally exhibit low genetic diversity compared to introduced species [19]. *C. ternatea* is believed to have originated from Ternate, before being introduced to Tidore, Halmahera, and Morotai due to its use as a herbal medicine and ornamental plant [20].

Butterfly pea is highly adaptable, thriving across various environmental conditions. This study examined the characteristics of the habitat influencing accession distribution including soil type, temperature, pH, altitude, side elevation, and land use. The mean annual rainfall in Ternate, Tidore, and Morotai was 178.8 and 197.8 mm$^3$. The plant is used in plantation areas and forests, as an ornamental yard flora, or as a land cover [14]. It can grow in various soil types and conditions, such as sandy, clay, and calcareous soil. Furthermore, the plant tolerates habitats flooded with high rainfall and arid, nutrient-poor soils. Butterfly pea can grow in a pH range of 5.0–8.9 and a temperature of 25–35 °C [15,16].

Soil, one of the abiotic components, plays a crucial role in supporting plant growth. According to previous reports, butterfly pea can grow across poor to nutrient-rich soil. Inceptisols, which are relatively nutrient-poor and categorized as young soils, are more prevalent compared to Molysols and Andosols, which are classified as fertile [17]. These soils were more predominant on Ternate than on Tidore and Morotai, with the Andisol soil on Tidore stemming from volcanic activities. The plant can grow in various soil conditions, supporting its spread in North Maluku.

Molecular genetic diversity analysis was used to describe the phenotypic diversity of butterfly pea [7]. Genetic diversity among accessions and their relationships were assessed using ISSR markers with fragment sizes ranging between 100 and 3000 bp. These markers are located adjacent to the microsatellites in the core genome region and can be amplified with the PCR technique. The microsatellite core sequences encompass primers, with a few selected nucleotides as anchors into the adjacent nonrepeating DNA region. The advantages of this technique include not requiring sequence data in designing primers,

minimal DNA for analysis purposes, 10–60-fragment amplification lengths, and targeting microsatellite regions in the core genome [21].

The molecular genetic diversity of butterfly pea assessed using ISSR markers showed that all accessions were grouped according to their similarity. All accessions in the population exhibited diversity by forming a single group at a genetic similarity level of 0.709. The variations occurred due to the diverse origin of the accessions from Ternate, Tidore, Morotai, and Halmahera. The results showed differences between ISSR profiles of the TTE 03 accession from group A and HAL 01 in B. Differences were also observed between MOR 01 from group A and HAL 01 in B within the same species. This suggests the potential of an ISSR primers' locus to distinguish different genotypes from the same species. The UPGMA grouping analysis showed that all accessions indicated no diversity at the 1.00 similarity level, signifying substantial genetic diversity, both morphologically and molecularly.

The frequency of alleles' appearance serves as a measure of polymorphic loci, determining the level of information produced by a DNA marker [22]. Polymorphic loci occur when the occurrence frequency is less than 0.99, meaning that one or more alleles can appear in one accession but not in another [23]. Polymorphic or allele diversity resulting from a marker describes the variation in each gene [24]. Genetic diversity is necessary for plant breeding and indicates high adaptability in the population [25]. This study also challenged the ability of the selective primers to detect the genetic variability in Indonesian butterfly pea.

Quantitatively, the polymorphism of a DNA marker can be determined by the PIC value [26]. ISSR was used as the dominant marker, and the UBC 889 primer PIC value applied to the Indian butterfly pea population exhibited the largest score of 0.55 [13]. Among 10 sets of the same primers, (AG)8C had the largest PIC value of 0.54. The same primer sets were used on the plant, resulting in varied PIC values at different locations. The number of polymorphic bands was very high compared to previous studies that used similar DNA markers. Although markers such as RAPD and ISSR have been utilized, the comprehensive documentation of DNA marker selection for butterfly pea genetic diversity analysis is lacking [9,10,13]. A recent study on genetic diversity among 14 genotypes of *Cajanus* spp., in India, showed a moderate variation of 55.1% using SSR primers [27].

Environmental heterogeneity is one of the ecological factors that greatly influence the genetic diversity of a population [28]. According to a previous study, genetic variation is related to habitat characteristics and species distribution patterns [29]. Abiotic and biotic components in an environment also trigger epigenetic mechanisms that influence plant genetic diversity [30]. This leads to differences in biochemical, physiological, and morphological aspects within a population. In this study, the dendrogram appearance with morphological and molecular markers for all accessions provided information on the high genetic diversity of butterfly pea populations in North Maluku.

The geographical location of the archipelago significantly affected genetic diversity, but the distribution of plants was more widespread on Ternate and Tidore compared to Morotai. Furthermore, the species distribution did not correlate with accession grouping in determining genetic diversity. The accession of butterfly pea on the four islands formed an interconnected group. The assessment of genetic similarity can be employed in identifying desired parents in future breeding programs. Butterfly pea flower has great potential as a native plant, which contributes to its wide distribution and diversity.

## 5. Conclusions

In conclusion, this study is the first report to assess genetic variation in several accessions of *C. ternatea* scattered across four islands in North Maluku with ISSR markers. The results showed very high genetic diversity with a similarity (monomorphic bands) and diversity (polymorphic band) value of 9% and 91% respectively, while the similarity index range based on the Jaccard coefficient was 60–92%. UPGMA tree analysis indicated a combination of accessions from Ternate, Tidore, Halmahera, and Morotai. This implied that *C. ternatea* from Ternate spread across all groups, potentially indicating its role as the oldest

and most adaptive accession across environments. This adaptability was more influenced by geographical location than differences in flower colors.

Knowledge of genetic diversity is vital for the efficient management of *C. ternatea* germplasm and for designing successful breeding programs. It will also guide the breeder on the most appropriate breeding strategy and mating design to maximize genetic gain per breeding cycle. Furthermore, adequate genetic diversity knowledge facilitates the efficient allocation of breeding resources and effective time management.

This study provides basic insights into the prospect of additional genetic marker investigation and detailed genomic studies in the future. For example, the establishment of a reference genome map can lead to more favorable efforts aimed at bridging the gap between genetic marker development and the use of gene-editing techniques in the future.

**Author Contributions:** Conceptualization, N.; methodology, N., A.K. and S.; formal analysis, N., A.K. and S.; original drafting, N.; review—writing, and editing, N., A.K., R.H., V.C. and T.S.; A.K. and V.C., funding acquisitions. All authors have read and agreed to the published version of the manuscript.

**Funding:** This study is part of the Ph.D. program at Universitas Padjadjaran, Bandung, Indonesia, supported by the Sensient Project in Indonesia, and funded by Universitas Padjadjaran (UNPAD).

**Data Availability Statement:** The datasets generated are available on request to the correspondence author.

**Acknowledgments:** The authors are grateful to the local community for information on existing plants in the study areas.

**Conflicts of Interest:** The authors declare that there is no conflict of interest.

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
