# Peer review of "The First Report on the Application of ISSR Markers in Genetic Variance Detection among Butterfly Pea (Clitoria ternatea L.) Accession in North Maluku Province, Indonesia"

_horticulturae, doi:10.3390/horticulturae9091059_

Round 1
Reviewer 1 Report
The manuscript entitled "The first report of application of ISSR markers in genetic vari- 2 ance detectionamong Butterfly Pea (Clitoria ternatea L.) Acces- 3 sion in North Maluku, Province, Indonesia" is written well. Howevers , there are several issues, which are need to addressed
A) The quality of Figures 1 and 4 are not good. Please replace figure 1 and 4 with good quality of figures.
b). The conclusion is too long. Please improve the conclusion and provide logical facts in conclusion.
c). Please provide the most suitable words
d). I have found many gramar mistakes and typing errors throughout the manuscript. Authors do not pay much attention on use of prepositions, use of articles, form of verbs and so on
for examples:
1. Please check title "The first report of application of ISSR markers in genetic vari- 2 ance detectionamong Butterfly Pea (Clitoria ternatea L.) Acces- 3 sion in North Maluku, Province, Indonesia".
I have found mistake in title. Please add the space between " dectection" and "among"
2. Butterfly Pea or butterfly pea? please make sure
3.inIndonesia. Please add the space
4.dye.The add space
5.and the scientific ranking hierarchy 50 has been reported in previous research [2][3][4][5]. Please ensure the format of reference
6.butterfly pea.For example. Typing errors
7. "is still limited.Currently, " same problem here
8."Ternate, Tidore,Halmahera and Morotai, from" typing error
9."plant species [14]The objective of" full stop missing
10."shown in Fig 1," is format correct?
11. "Sampling of Plant material". please make sure the format of writing should be same. I have found several these kind of typing mistakes.
12. "accessionscollected from " add space
13."performed using The Wizard® Genomic DNA Purification Kit" . Please replace "The" with "the"
14."All accessions showed a genetic similarity level of 0.709, hence butterfly pea 30 flower from North Maluku has a high degree of genetic diversity". Flowers
15. Add space before it . "Butterfly pea (Clitoria ternatea L.) is a leguminous plant from the Fabaceae family.It is"
16. Pea or peas?
17. Add space before Currently. "including Maluku, is still limited.Currently, flower morphology characters".
18. Please correct as " groups" . "which is based on the average linkage between group, i.e. the"
19. Please correct the spelling of (dendogram) "The accessions of C ternatea are spread over all subgroups on the dendogram,"
20. "home gardeners which may result in genetic lineage similarity[13] However, the four." Please Put Punctuation before "However".
21. "This study also gives challenge to the selective primers to detect genetic variabilty of Indonesian butterfly peas". Please correct the spelling "variabilty "
22. and many others
I have found many gramar mistakes and typing errors throughout the manuscript. Authors do not pay much attention on use of prepositions, use of articles, form of verbs and others
Author Response
Dear Reviewer 1
We Hereby sent respon to reviewer 1 for suggestion for revising the article an title First report of application of ISSR markers in genetic variance detection among Butterfly Pea (Clitoria ternatea L.) Accesion in North Maluku, Province, Indonesia.
Best Regard
Sundari

Reviewer 2 Report
The paper horticulturae-2554518 describes the sampling of 18 accessions of Butterfly pea from four islands (Ternate, Tidore, Morotai, and Halmahera) at North Maluku. The 18 accessions were analysed with ten ISSR markers, which do not require previous knowledge of the genome sequencing. The amplification pattern revealed a good number of polymorphic bands per primer and PIC values varying from 0.21 to 0.54. In general, authors conclude that Butterfly pea from North Maluku shows high genetic variability.
Personally, I think that any information regarding orphan plant species is valid. Although the paper is quite simple and uses a modest molecular tool, it shines some light in the diversity of Butterfly pea. Clearly, it paves the way for new questions, as: (1) What is driven the high genetic diversity in Butterfly pea? (2) Can the diversity of Butterfly pea be fully explored to improve important traits? (3) Is the diversity greater in one of the islands and why? In that context, the simplicity of the paper can be seen as a first step towards more complex questions.
My comments/suggestions are:
(1) Lines 51-52: “Additional data on genetic diversity can be interesting because it can contribute to better understanding of the degree of diversity in butterfly pea”. This sentence seems redundant. Please, give other answers on why additional data on genetic diversity is interesting.
(2) In the Introduction, add information on the availability of genome resources of Butterfly pea. Is the genome being sequenced? Are there any other DNA sequencies deposited in the GenBank?
(3) Line97: In the legend of Figure 1, is it “research area” or “area of sampling”?
(4) Is there any information about the location where the plants were collected? I mean, what is the type of soil, presence of pathogens/pests, etc?
(5) Are the characteristics (type of soil, presence of pathogens/pests, UV radiation, temperature over the year, rain fall over the year) of the islands similar or significantly different?
(6) Lines110-112: Please indicate in the figure and in the legend which picture represents each island (just include A, B, C, and D in the pictures and indicate which islands are represented by A, B, C, and D).
(7) Line 128: Please indicate the annealing temperature.
(8) Lines 153-154: Add “each of” after “by”. It should be: “The total bands produced by each of the 10 ISSR primers …”
(9) Table 2: It will be interesting to list the primers based on the PIC values (higher to lower).
(10) Line 164: “eggplant”?
(11) Line 168: Replace “Figure 4” by “Table2”.
(12) Line 169: In the end of the sentence, add “(Figure 4). It should be: “Cluster analysis of 18 butterfly pea accessions formed groups A and B (Figure 4)”.
(13) The discussion and conclusion are too long. Please, be more concise.
(14) The discussion should include the following points:
(14.1) What about the genetic variability within each island? If there are differences, why the genetic variability is greater in on island compared to other(s)?
(14.2) Lines 223-225: “C. ternatea probably originated from the island of Ternate, then was introduced to the islands of Tidore, Halmahera and Morotai due to the use of C ternatea as a herbal medicine and ornamental plant and then underwent naturalization [21]”. Is it possible to argue against or in favour of this theory based on the genetic profile of Butterfly pea from each island?
(14.3) Botstein et al. (Am J Hum Genet. 32:314–331, 1980) proposed that PIC> 0.50 indicates highly informative markers, PIC from 0.25 to 0.50 indicates reasonably informative markers, and PIC < 0.25 represents slightly informative markers. Is it possible to use that information in the ISSR primers used here? In that case, only marker (AG)8C will be considered highly polymorphic. Why only one in ten ISSR markers is highly polymorphic?
Author Response
Dear Reviuwer 2
Hereby We sent respons of comment from reviewer 2 on manuscript an tittle The First Report on the Application of ISSR Markers in Genetic Variance Detection Among Butterfly Pea (Clitoria Ternatea L.) Accession in North Maluku Province, Indonesia.
Best Regard
Sundari

Reviewer 3 Report
Dear Authors,
The article is not written in scientific language. It is very difficult to understand the text.
The title of the manuscript is misspelled.
The annotation structure is incorrect. Look at the already published articles in order to properly format it.
Introduction:
line 51: format links like [2-5]
line 86: no dot after [14]
line 63,64,72,93: numbers less than 10 write in words
Materials and methods:
The structure of this section is not well thought out. Sections 2.1 and 2.2.1 carry the same information.
- Figure 1 is of very poor quality, plant sampling points are not indicated
- Figure 2 does not carry any information
- lines 121-122 and table 2: Maybe SSR markers, not primers?
-2.2.4 does not describe how the alleles of SSR markers were detected.
There is very little information in the results, there is a listing of information from tables and figures without their understanding.
164 line: your object of research is eggplant?
There is a lot of unnecessary information in the discussion that is not related to this work. Somewhere the dots are superfluous, somewhere they are missed. "dendrogram" but not "dendrogram".
The bibliography is not formatted according to the rules of the journal.
I did not list all the comments, since the work needs to be completely redone.
"dendrogram" but not "dendrogram".
Verb tenses are incorrect.
Author Response
Hereby We sent respons of comment from reviewer 3 on manuscript an tittle The First Report on the Application of ISSR Markers in Genetic Variance Detection Among Butterfly Pea (Clitoria Ternatea L.) Accession in North Maluku Province, Indonesia.
Best Regard
Sundari

Round 2
Reviewer 3 Report
Dear Authors,
After the revision, the manuscript has become much better.
I have a few small comments.
l. 43 extra dot after "... the Fabacea family. native..."
l. 61 Numbers less than 9 are written in words
l. 64 no space between "...ing.According..."
l. 157 Figure 3, not 4?
l. 162 Missing link to Figure 4
l. 184 "showcases" - there should be a Space.
Please, prepare the References in accordance with the requirements of the journal and uniformly. For example, links 3, 4 and 6 are completely different in design.
Author Response
Dear Reviuwer 3
I sent Revised for manuscript an tittle: The First Report on the Application of ISSR Markers in Genetic Variance Detection Among Butterfly Pea (Clitoria Ternatea L.) Accession in North Maluku Province, Indonesia. with add diferent colour in revised.Thanks you for good attention
Best regard
Prof.Sundari
